# Aflatoxin M_1_ Analysis in Urine of Mill Workers in Bangladesh: A Pilot Study

**DOI:** 10.3390/toxins16010045

**Published:** 2024-01-14

**Authors:** Nurshad Ali, Ahsan Habib, Firoz Mahmud, Humaira Rashid Tuba, Gisela H. Degen

**Affiliations:** 1Department of Biochemistry and Molecular Biology, Shahjalal University of Science and Technology, Sylhet 3114, Bangladesh; ahsan09bmb15@student.sust.edu (A.H.); firozmahmud393@gmail.com (F.M.); humairatuba39@gmail.com (H.R.T.); 2Leibniz-Research Centre for Working Environment and Human Factors (IfADo) at the TU Dortmund, Ardeystr. 67, D-44139 Dortmund, Germany

**Keywords:** aflatoxin, Bangladesh, ELISA, exposure, mill workers, urine

## Abstract

Presence of aflatoxin B_1_ (AFB_1_) in food and feed is a serious problem, especially in developing countries. Human exposure to this carcinogenic mycotoxin can occur through dietary intake, but also through inhalation or dermal contact when handling and processing AFB_1_-contaminated crops. A suitable biomarker of AFB_1_ exposure by all routes is the occurrence of its hydroxylated metabolite aflatoxin M_1_ (AFM_1_) in urine. To assess mycotoxin exposure in mill workers in Bangladesh, we analyzed AFM_1_ levels in urine samples of this population group who may encounter both dietary and occupational AFB_1_ exposure. In this pilot study, a total of 76 participants (51 mill workers and 25 controls) were enrolled from the Sylhet region of Bangladesh. Urine samples were collected from people who worked in rice, wheat, maize and spice mills and from controls with no occupational contact to these materials. A questionnaire was used to collect information on basic characteristics and normal food habits of all participants. Levels of AFM_1_ in the urine samples were determined by a competitive enzyme linked immunosorbent assay. AFM_1_ was detected in 96.1% of mill workers’ urine samples with a range of LOD (40) of 217.7 pg/mL and also in 92% of control subject’s urine samples with a range of LOD of 307.0 pg/mL). The mean level of AFM_1_ in mill workers’ urine (106.5 ± 35.0 pg/mL) was slightly lower than that of the control group (123.3 ± 52.4 pg/mL), whilst the mean AFM_1_ urinary level adjusted for creatinine was higher in mill workers (142.1 ± 126.1 pg/mg crea) than in the control group (98.5 ± 71.2 pg/mg crea). Yet, these differences in biomarker levels were not statistically significant. Slightly different mean urinary AFM_1_ levels were observed between maize mill, spice mill, rice mill, and wheat mill workers, yet biomarker values are based on a small number of individuals in these subgroups. No significant correlations were found between the study subjects’ urine AFM_1_ levels and their consumption of some staple food items, except for a significant correlation observed between urinary biomarker levels and consumption of groundnuts. In conclusion, this pilot study revealed the frequent presence of AFM_1_ in the urine of mill workers in Bangladesh and those of concurrent controls with dietary AFB_1_ exposure only. The absence of a statistical difference in mean biomarker levels for workers and controls suggests that in the specific setting, no extra occupational exposure occurred. Yet, the high prevalence of non-negligible AFM_1_ levels in the collected urines encourage further studies in Bangladesh regarding aflatoxin exposure.

## 1. Introduction

Cereal grains play an important role in both the human diet and in livestock feed due to their valuable nutrient contents, such as proteins, carbohydrates, fatty acids, vitamins and minerals [1,2]. However, these agricultural products can be contaminated by various fungi in the field or during harvest and storage. Fungal infestation and mycotoxin production in crops depend on storage conditions, climate, temperature, insects and drought [3,4].

Mycotoxin contamination of cereals and grain-based food products is a significant global issue, with aflatoxins, produced by fungi of the *Aspergillus* genus, being the most toxic contaminants in a large portion of the world’s food supply [5,6,7]. Humans are exposed to aflatoxins through contaminated dietary staples, including maize, peanuts, rice and various cereal-based products [8,9,10]. Regions in Africa and Southeast Asia with climatic conditions favorable for fungal growth often experience greater contamination of feed and food due to poor storage conditions for crops [10,11].

It is well known that aflatoxins are harmful to animals and humans, causing severe toxicities at acute high or chronic low doses [12,13,14]. Aflatoxin B_1_ (AFB_1_) is the most potent of the aflatoxins, all classified by the IARC as human carcinogens [15]. There are strong correlations between chronic AFB_1_ exposure and the risk of developing hepatocellular carcinoma, the third leading cause of cancer deaths worldwide [16]. Thus, regulations exist on maximal levels of aflatoxins, including AFB_1_, AFG_2_, AFB_2_ and AFG_2_, for food crops and for AFM_1_ in milk [6,17]. Bangladesh now also has specific regulatory limits for contamination of certain food items [18].

AFB_1_ and its hydroxylated metabolite AFM_1_ have similar toxic properties, including carcinogenic and hepatotoxic effects [17,19,20]. Ingested AFB_1_ is partly converted in the organism to AFM_1_ and then excreted with urine and breast milk, commonly used matrices for analysis in human biomonitoring studies, with urinary AFM_1_ serving as a valid short-term biomarker of AFB_1_ exposure [21,22].

Biomonitoring is an effective method of assessing human exposure, as illustrated by results of earlier studies conducted in various populations that indicate common dietary exposure to major mycotoxins, albeit at different levels [23,24,25]. Analysis of suitable biomarkers in human body fluids provides useful insights into dietary exposure when food contaminant data are sparse or insufficient, as is often the case in developing countries [8,26,27]. Furthermore, toxigenic fungi and mycotoxins present in occupational and residential settings may lead to an extra exposure by inhalation or by dermal contact [28,29]. During the handling and processing of contaminated crops, spores and small particles of raw materials can give rise to mycotoxin exposures through organic dusts at certain workplaces [30,31,32]. Therefore, it is worthwhile to investigate whether workplace-related mycotoxin exposures will add significantly to those resulting from oral intake of contaminated food by analyzing mycotoxins and/or metabolites in biofluids of workers in ’risky’ settings and comparing the levels with those of control subjects [28,33,34]. A systematic review of studies with such a design, using a biomarker analysis of blood or urine samples, revealed that feed mill workers had the greatest exposure to mycotoxins, especially to aflatoxins [32]. This may be due to the usually poorer quality of raw materials for animal feeds than grains processed for human food.

In Bangladesh, data on AFB_1_ contamination of foods and feeds are sparse. An early survey found variable incidences of contamination in different commodities (e.g., 8% in rice and 67% in maize) and notably higher AFB_1_ levels in maize, groundnuts and poultry feed than in rice and pulses [35]. Aflatoxin analysis of crop samples from six districts of Bangladesh revealed also variable incidence rates and levels for maize, wheat and rice, with the highest values for maize [36]. A study of eight food commodities (rice, lentils, wheat flour, dates, betelnut, red chili powder, ginger, groundnuts) reported the highest aflatoxin levels in dates and groundnuts, and concentrations exceeded the US regulatory maximum regulatory limits in five of the eight commonly ingested food commodities tested [37]. In line with such findings, our biomonitoring studies on occurrence of AFM_1_ in the urine of adults and pregnant women and in milk of nursing mothers document widespread exposure to the mycotoxin AFB_1_ in the general population [8,38,39]. But there is currently no such data available for mill workers. Therefore, this study has determined the presence of AFM_1_ in the urine samples of cereal grain and spice mill workers in the Sylhet region in Bangladesh and compared the results with those obtained for control subjects in the same region. Correlating biomarker data with information collected on food habits aimed to identify major sources of AFB_1_ intake. 

## 2. Results

### 2.1. Characteristics of the Study Groups

The group of mill workers comprises employees of three grain mills where rice, wheat and maize are processed and one spice mill (for chili peppers, turmeric, garam masala). The control group consisted of individuals recruited from the same region, including day laborers in other professions, rickshaw pullers, housewives, teachers, and businessmen (Table 1). Among the total of 76 participants, 51 were mill workers (37 men and 14 women), and 25 were controls (17 men and 8 women). 

The average age was 38.76 years for mill workers and 38.64 years for the control group. The control group had a higher mean body mass index (BMI, 24.94 ± 3.45 kg/m^2^) compared to the mill workers (21.74 ± 3.07 kg/m^2^) (*p* < 0.001). This may be a chance finding as BMI ranges indicate that in both groups there were underweight and overweight people. There was no significant difference in the mean level of urinary creatinine between the mill workers (1.34 ± 0.91 mg/mL) and control group (1.78 ± 1.07 mg/mL). The majority of participants in both groups belonged to the low socioeconomic status group (82.4% vs. 88%) and had a primary or elementary level of education (over 70%).

### 2.2. Levels of AFM_1_ in Urine of Mill Workers and Control Group

The detection frequency, levels and distribution of AFM_1_ in mill workers and controls are presented in Table 2 and Figure 1. AFM_1_ was found in 96.1% of mill workers’ urine samples with a range of LOD of 217.7 pg/mL and also in 92% of control group urine with a range of LOD of 307.0 pg/mL. The mean urinary AFM_1_ concentration in mill workers (106.5 ± 35.0 pg/mL) was slightly lower than in the control group (123.3 ± 52.4 pg/mL), whereas the mean AFM_1_ urinary level adjusted for creatinine was higher in mill workers (142.1 ± 126.1 pg/mg crea) than in the control group (98.5 ± 71.2 pg/mg crea). These and differences in the median AFM_1_ levels between groups did not reach statistical significance. Also, gender had no effect on AFM_1_ and creatinine-adjusted AFM_1_ levels, which is not surprising in light of the inter-individual variability in both groups (Figure 1).

In Table 3, the mill workers were grouped according to the different materials processed at the mills. All participants working in maize and spice mills had measurable levels of AFM_1_ in their urine (100%); % positive detects in rice mill (95%) and wheat mill (89%) workers were a bit lower. The mean AFM_1_ urine concentration was higher in workers from spice (116.3 ± 40.4 pg/mL) and maize (115.7 ± 26.4 pg/mL) mills compared to those working in rice (102.4 ± 32.7 pg/mL) and in wheat (92.6 ± 41.2 pg/mL) mills. After adjusting for creatinine, mean AFM_1_ levels were found to be higher in spice (205.0 ± 192.9 pg/mg crea) and in rice (146.5 ± 123.4 pg/mg crea) mill workers than in the other workers’ urine. Yet, it has to be noted that this comparison is based on a small number of subjects.

### 2.3. Relationship between AFM_1_ Biomarker Levels and Food Consumption

We classified all participants based on information in the questionnaires for ’regular’ rice consumption and rice consumption in the 2 days prior to urine collection (Table 4). We found that those who consumed rice more often (3 times/day or 5–6 times in 2 days) had higher mean AFM_1_ urine levels than those who consumed rice less frequently (1–2 times/day or 2–4 times in 2 days), a difference only significant for crea-adjusted AFM_1_ (*p* < 0.05) in the last two days before sampling. Then, possible correlations were assessed between individual food consumption patterns (regular and two days prior to sample collection) and the AFM_1_ levels found in urine samples. We compared consumption frequencies of major food items such as rice, wheat/maize, milk, pulses and groundnuts using Spearman correlation analysis (Table 5). The analysis showed a significant correlation between AFM_1_ levels and consumption of ground nuts in the regular and last two days food consumption groups (*p* < 0.01 and *p* < 0.05, respectively). Yet, no significant correlations were found between AFM_1_ levels and consumption frequency of rice, wheat/maize, milk and pulses. 

## 3. Discussion

As outlined in the Introduction, dietary exposure to mycotoxins, in particular aflatoxins, is an issue of significant concern due to the widespread contamination of major food and feed commodities [6,9,11]. Moreover, there is also growing interest in the role of mycotoxins as health hazards in occupational settings, e.g., agricultural and food processing facilities, or the waste management sector [28,40]. The general population ingests mycotoxins mainly with contaminated foods, whilst workers in certain settings may have additional exposure by inhalation of mycotoxins with organic dusts when handling crops intended for human consumption or other materials such as animal feed [30]. Analysis of settled dust and ambient air monitoring attest the presence of toxigenic fungi and major mycotoxins, including aflatoxins, at various workplaces (references in [31,32]. Yet, assessing risks from workplace-related mycotoxin exposures is a challenging task due to uncertainties regarding the possible impact of a respiratory intake and/or by dermal contact [28]. To reduce these uncertainties, biomonitoring has been applied to investigate and compare biomarker levels in human fluids obtained from workers and from controls, i.e., non-occupationally exposed people. So far, studies on aflatoxins have used biomarker analysis (free or albumin-bound AFB_1_) of blood serum samples, reviewed in [31,32], whilst others have analyzed AFM_1_ in urine samples [34,41,42,43]. Studies conducted in different settings reported a higher proportion of positive samples and/or higher biomarker levels in workers than in controls, whilst others did not find a significant group difference. 

The present biomarker-based study is the first one to investigate workplace-related exposure to aflatoxins in Bangladesh, namely in employees of grain and spice mills from the Sylhet city region, and in a local control group with only dietary mycotoxin exposure. The study results did not reveal significant differences in AFM_1_ biomarker concentrations between the mill workers and control subjects (Table 2). On the other hand, urine of the mill worker groups contained higher mean and maximal levels of crea-adjusted AFM_1_, indicative of a possible workplace exposure. When mill workers were grouped by the material processed (Table 3), workers of the spice mill had the highest average level of AFM_1_ (116.3 ± 40.4 pg/mL or 205.0 ± 192.9 pg/mg crea); workers of wheat mills presented the lowest biomarker levels (92.6 ± 41.2 pg/mL or 77.9 ± 26.0 pg/mg crea). Of note, studies in some Asian countries have reported the presence of aflatoxins in different spices [44,45,46,47], yet in Bangladesh, data on AFB_1_ contamination of spices and other food commodities are scarce. 

Recent studies by others on workplace-related exposure to aflatoxins in Europe found overall clearly lower AFM_1_ biomarker levels than in our Bangladeshi mill workers. A study conducted in Italy measured a mean urinary AFM_1_ concentration of 35 pg/mL in feed mill workers and 27 pg/mL in controls [41]. A follow-up study in the same feedstuff plant with a refined analytical method for samples collected later on, detected AFM_1_ concentrations ranging from 1.9 to 10.5 pg/mL in 13% of the workers’ urine, yet they were not significantly different from values in concurrent controls [42]. A pilot study in France of nine workers during cleaning of a grain elevator detected AFM_1_ in four of nine urine samples at a mean level of 316 pg/mL and 413 pg/mg crea, indicative of an occupational exposure, which was probably due to rather high levels of AFB_1_ found in airborne dust samples [43]. 

The present biomarker data on mycotoxin exposure for our cohort from the Sylhet district may be further compared to results of a previous study conducted in Bangladesh which also used the ELISA technique to detect AFM_1_ in urine samples collected from rural and urban residents of the Rajshahi district [38]. In that study, AFM_1_ was detected in 46% of urine samples, at concentrations ranging from 31 to 348 pg/mL, with a mean AFM_1_ level of 80 ± 60 pg/mL and a crea-adjusted mean value of 130 ± 90 pg/mg crea. Two other studies in Bangladesh which used HPLC-FD reported similar and lower AFM_1_ levels in the urine of adults, a pregnant women cohort and in children that indicate widespread dietary mycotoxin exposure [8,26]. Considering this, we looked for a possible correlation between the individual biomarker levels and the consumption frequency of certain food categories in the present cohort. We found no significant correlation between urinary biomarker levels and the consumption of main staple food items ingested two days prior to urine sampling and regular food habits. However, we observed a significant correlation between the AFM_1_ biomarker levels and the consumption of groundnut, a finding in accord with previous reports on aflatoxin contamination of groundnuts in Bangladesh [35,37]. When rice consumption was categorized as low or high, higher levels of crea-adjusted mean AFM_1_ were observed in the high consumption group. Rice, being a primary food source in many Asian countries, can be contaminated with aflatoxins [48]. But rice is commonly consumed with curries prepared with various spices which may also be a source of AFB_1_ intake in the Bangladeshi population [8] since mycotoxin contamination of spices is known to occur in several Asian countries [44,45,46,47].

Our study had some limitations: There was evidence for a dusty environment at the workplaces, yet due to limited resources, we could not establish ambient monitoring for mycotoxins. We also could not collect information on occupational exposure variables (e.g., workspace volume, ventilation rate, humidity, temperature and working hours) that may affect the biomarker concentration. Furthermore, we used ELISA for the AFM_1_ analysis, which is less sensitive than HPLC-FD or LC-MS/MS analysis and may lead to a slight overestimation of biomarker values. Yet, a strength of our study is that we have detected AFM_1_ levels in a high percentage of both workers and control subjects. Even with a small number of participants, the biomarker results showed that both groups encountered widespread AFB_1_ exposure.

## 4. Conclusions

This pilot study revealed frequent presence of AFM_1_ in the urine of mill workers in Bangladesh and those of concurrent controls with dietary AFB_1_ exposure only. The absence of a statistical difference in mean biomarker levels for workers and controls suggests that in the specific setting, no extra occupational exposure occurred. Yet, the high prevalence of non-negligible AFM_1_ levels in the collected urine should trigger further biomonitoring studies at workplaces in the food and feed industry of Bangladesh. 

## 5. Materials and Methods

### 5.1. Study Subjects and Sampling

A total of 76 participants were enrolled for this study, of which 51 were mill workers and 25 were healthy controls. The study was conducted in the Sylhet city region of Bangladesh between June 2021 and December 2021. The mill workers were selected from three different grain mills, including rice, wheat and maize mills, as well as a mill for spices (chili peppers, turmeric, mixed masala), where such items are processed for human food. The healthy control subjects were housewives, daily labor-based workers, rickshaw pullers, businessmen, and other people from the same region with no occupational contact to grains. Each participant provided about 50 mL morning urine sample in a non-sterile disposable container which was stored at −20 °C at the Department of Biochemistry and Molecular Biology of Shahjalal University of Science and Technology, Bangladesh. Urinary creatinine concentrations were measured to account for differences in urine dilution between individual spot urines [49]. Urine creatinine was measured with a colorimetric method according to the protocol provided by the manufacturer (HUMAN Gesellschaft für Biochemica und Diagnostica mbH, Wiesbaden, Germany) using a semi-automatic biochemistry analyzer (Humalyzer 3000, Medicon Services, Tuttlingen, Germany). All participants were asked to fill out a questionnaire providing anthropometric and demographic information as well as their food consumption habits. Written consent was obtained from all participants before they were included in the study. This study was approved by the Internal Ethics Review Board at the Department of Biochemistry and Molecular Biology, School of Life Sciences, Shahjalal University of Science and Technology, Sylhet, Bangladesh (Reference number: 01/BMB/2020). 

### 5.2. Food Consumption Data 

The food frequency questionnaire (FFQ) included questions on regular food consumption habits, as well as what participants had eaten in the past two days. It listed common food items typically consumed by the Bangladeshi population, such as rice, wheat, maize, pulses, milk and ground nuts. The questionnaire did not ask for information about the consumption of spices. The frequency of food consumption was graded 0 to 3, as shown in the footnote of Table 4. The only food item that the majority of participants consumed up to three times a day was rice.

### 5.3. Sample Preparation and AFM_1_ Analysis

Urinary AFM_1_ levels were measured using an enzyme-linked immunosorbent assay (ELISA) as described elsewhere [38]. The ELISA kits for aflatoxin M_1_ (Catalog #991AFLM01U-96) were purchased from Helica Biosystems Inc., Santa Ana, CA 92704, USA. The urine samples were centrifuged at 3200× *g* for 5 min, and the supernatant was used for AFM_1_ determination following the procedure specified in the method protocol. To summarize the procedure, both the AFM_1_ standards and urine samples were diluted with distilled water (1:20 *v*/*v*), and 100 μL of each was mixed with 200 μL assay buffer. Then, 100 μL of this mixture was transferred to an antibody-coated microtiter well, and the plate was incubated at room temperature (RT) for 1 h. The plate was washed with the wash buffer using an automated microplate washer (Wellwash™ Microplate Washer, Thermo Scientific, Waltham, MA, USA). In each well, 100 μL of AFM_1_ conjugate was added and incubated at RT for 15 min. After that, the plate was washed to remove the unbound conjugate. Then, 100 μL of substrate reagent was added to each well, and the color reaction was allowed to proceed for 15 min in the dark at RT. Later, 100 μL of stop solution was added to the wells to terminate the enzyme reaction. Within 15 min, absorbance was measured at 450 nm in a microplate reader (Apollo 11 LB 913, Berthold Technologies, Bad Wildbad, Germany). The absorption intensity is inversely proportional to the concentration of AFM_1_ in the samples. The level of AFM_1_ in the samples was calculated from the concurrent standard curves. To validate the method, two different concentrations (60 and 100 pg/mL) of AFM_1_ standard were added to blank urine samples, and the recovery rate was found to be in the range of 85–105%. The method detection limit (LOD) was determined to be 40 pg/mL. 

### 5.4. Statistical Analysis

The data were analyzed using IBM SPSS Statistics version 22. Descriptive analysis was performed to determine the mean, median and interquartile range of the analyte. The baseline characteristics and analyte concentration differences between the mill workers and control group or gender were analyzed using an independent sample *t*-test. Chi-square test was applied for categorical variables. AFM_1_ levels within the mill worker groups were compared using ANOVA test. The Spearman correlation coefficient (two-tailed) was used to assess correlations between food consumption and urinary AFM_1_ concentration. A statistical significance level of alpha *p* < 0.05 was assigned.

## Figures and Tables

**Figure 1 toxins-16-00045-f001:**
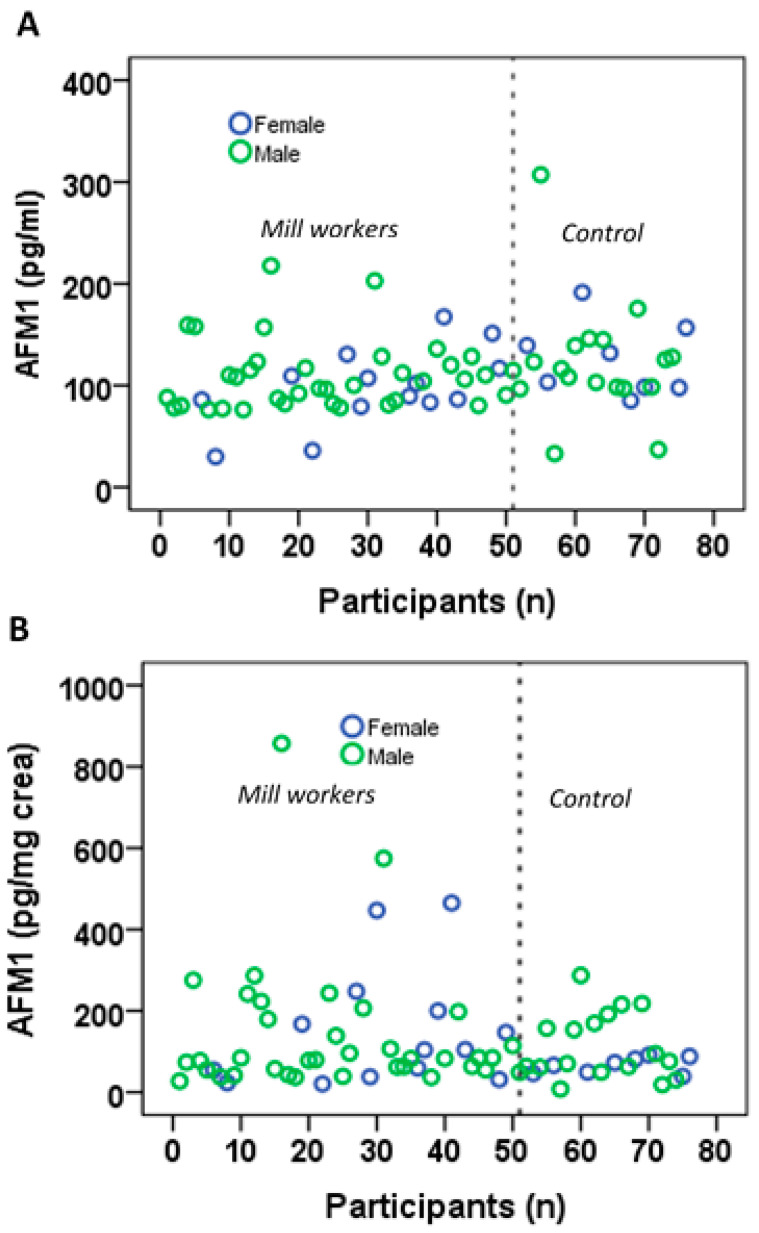
Distribution of normal (**A**) and creatinine-adjusted (**B**) AFM_1_ in urine of the participants (*n* 1–51 are mill workers, and *n* 52–76 are control subjects).

**Table 1 toxins-16-00045-t001:** Baseline characteristics of the participants.

Variables	Mill Workers	Control Group	*p*-Value
Subjects (*n*)	51	25	-
Mill-based workers (*n*)			
Rice	20	-	-
Wheat	9	-	-
Maize	11	-	-
Spices	11		
Gender (m/f)	37/14	17/8	0.439
Age (years)	38.76 ± 10.92	38.64 ± 10.21	0.961
BMI (kg/m^2^)			
Mean ± SD	21.74 ± 3.07	24.94 ± 3.45	<0.001
Range	16.69–30.47	19.38–29.76	-
Creatinine (mg/mL)	1.34 ± 0.91	1.78 ± 1.07	0.086
Socioeconomic status (*n*, %)			0.170
Low	42 (82.4)	22 (88.0)	
Medium	8 (15.7)	1 (4.0)	
Upper medium	1 (2.0)	2 (8.0)	
Education level (*n*, %)			0.347
Primary	37 (72.5)	18 (72.0)	
Secondary	12 (23.5)	4 (16.0)	
Above secondary	2 (3.9)	3 (12.0)	

Values are presented as mean ± standard deviation or *n* (%). *p*-values are obtained from independent sample *t*-test for continuous variable and chi square test for categorical variables.

**Table 2 toxins-16-00045-t002:** Occurrence and levels of AFM_1_ in the mill workers and control groups.

	Group	*n*	Positive *n* (%)	Mean ± SD		Median		Maximum	
				pg/mL	pg/mg Crea	pg/mL	pg/mg Crea	pg/mL	pg/mg Crea
Mill workers	Male	37	37 (100.0)	109.7 ± 33.9	138.8 ± 131.3	104.1	82.5	217.7	857.1
	Female	14	12 (85.7)	98.2 ± 37.9	150.8 ± 126.9	95.7	104.3	167.4	465.1
	Total	51	49 (96.1)	106.5 ± 35.0	142.1 ± 126.1	102.0	83.1	217.7	857.1
Controls	Male	17	15 (88.2)	122.2 ± 59.5	113.3 ± 81.8	116.4	76.2	307.0	287.3
	Female	8	8 (100.0)	125.5 ± 36.2	66.8 ± 20.1	117.5	69.4	191.4	91.6
	Total	25	23 (92.0)	123.3 ± 52.4	98.5 ± 71.2	116.4	72.6	307.0	287.3

Positive samples refer to urine containing the analyte ≥ limit of detection (LOD: 40 pg/mL). Samples below LOD were assigned half of LOD during calculation of mean and median values.

**Table 3 toxins-16-00045-t003:** Occurrence and levels of AFM_1_ in urine of different subgroups of mill workers.

Samples	*n*	Positive *n* (%)	Mean ± SD		Median		Max	
			pg/mL	pg/mg Crea	pg/mL	pg/mg Crea	pg/mL	pg/mg Crea
Rice mill workers	20	19 (95.0)	102.4 ± 32.7	146.5 ± 123.4	98.7	89.6	202.8	574.5
Wheat mill workers	9	8 (88.9)	92.6 ± 41.2	73.9 ± 57.9	80.2	54.6	159.3	275.5
Maize mill workers	11	11 (100.0)	115.7 ± 26.4	127.0 ± 111.8	114.7	84.9	167.4	465.1
Spices mill workers	11	11 (100.0)	116.3 ± 40.4	205.0 ± 192.9	109.3	167.7	217.7	857.1

Positive samples refer to urine containing the analyte ≥ limit of detection (LOD: 40 pg/mL). Samples below LOD were assigned half of LOD during calculation of mean and median values.

**Table 4 toxins-16-00045-t004:** Urinary AFM_1_ biomarker levels and regular or last two days of rice consumption prior to urine donation of all study participants.

	N	AFM_1_ (pg/mL)		AFM_1_ (pg/mg Crea)	
		Mean ± SD	Max	Mean ± SD	Max
Regular rice consumption					
1–2 times/day	17	117.5 ± 26.0	159.3	103.0 ± 60.8	247.8
3 times/day	59	110.5 ± 45.6	307.0	134.9 ± 129.7	857.1
Last 2 days rice consumption					
2–4 times/2 days	17	114.7 ± 27.1	159.3	90.7 ± 49.7	198.0
5–6 times/2 days	59	111.3 ± 45.5	307.0	138.5 ± 129.9 *	857.1

* *p* < 0.05 when compared to 2–4 times/day in the last 2 days rice consumption group. *p*-value is derived from independent sample *t*-test.

**Table 5 toxins-16-00045-t005:** Correlation of urinary AFM_1_ concentrations and food consumption frequency ^#^.

Food Items	Regular Food Consumption Habits		Last 2 Days Food Consumption	
	Correlation (r)	*p*-Value	Correlation (r)	*p*-Value
Rice	0.035	0.762	0.038	0.725
Wheat/maize	0.144	0.214	0.186	0.108
Milks	0.136	0.243	0.208	0.071
Pulses	0.083	0.474	0.015	0.898
Ground nuts	0.304	0.008	0.259	0.024

^#^ Assessment of food consumption frequency was carried out using numerical scores for the following food items; for rice: 1 = 1 time/day, 2 = 2 times/day, 3 = 3 times/day; wheat/maize: 0 = 0 time/day, 1 = 1 time/day, 2 = 2 times/day; milk: 0 = 0 time/day, 1 = 1 time/day; pulses: 0 = 0 time/day, 1 = 1 time/day, 2 = 2 times/day; groundnut: 0 = 0 time/day, 1 = 1 time/day, 2 = 2 times/day. *p*-values are obtained from Spearman’s correlation coefficient (two-tailed).

## Data Availability

Data are available from the corresponding author upon reasonable request.

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
