# Peer review of "Aflatoxin M1 Analysis in Urine of Mill Workers in Bangladesh: A Pilot Study"

_toxins, 2024, doi:10.3390/toxins16010045_

Round 1

Reviewer 1 Report

Comments and Suggestions for Authors

The manuscript submitted to “Toxins” an MDPI journal entitled: “Aflatoxin M1 analysis in urines of mill workers in Bangladesh: A pilot study” which discussed Presence of aflatoxin B1 (AFB1) in food and feed is a serious problem, especially in developing countries. Human exposure to this carcinogenic mycotoxin can occur by dietary intake, but also by inhalation or dermal contact when handling and processing AFB1-contaminated crops, the following points should be followed:

-          Abstract:

·         Should be rewritten in more details and high lighting the main results in order to sound better and giving strength to the manuscript.

·         Examination of biomarkers should be mentioned firstly the mentioning their results.

-          Introduction:

·         Was written in organized manner but recent literatures are required.

-          Materials and methods:

·         Number of samples is very small to be representative, as well as within the 4 classes rice, wheat, maize and spices.

·         Samples from mill crops should be collected and examined for AFM1 monitoring, in order to determine the source of AFM1, parallel with urine samples of mill workers.

·         Samples from food consumed by mill workers and control group should also collected and examined for detection of AFM1 to harmonize the result exactly.

·         Why did you examined only AFM1 not also AFB1 too?

·         What is the period between the last exposure to mill crops and the urine samples collection, and was there continuous working or found holidays for intermittent exposure for some workers, please clarify.

·         The health status about the tested human was not studied, before examining the creatinine conc.

·         Many references were missed in this section.

·         Creatinine adjusted value should be mentioned here with more details.

·         Why in questionnaire consumption of spicy not discussed as other types of food?

-          Results:

·         Result was poorly written and should be rewritten with organized

·         Table 1: when did you perform the test of creatinine monitoring?

·         BMI: write the whole words first the abbreviate them.

·         Why the conc. of AFM1 in control group is higher than the examined ones???

·         The results of each type of crops should be mentioned separately to make comparison between them and to study the possible causes.

·         What are the possible sources in control group, as the control group AFM1 is higher than examined ones so the mill itself not the expected source as it is the core idea of the manuscript.

·         A more strange point that AFM1 in control group was higher than examined ones although the creatinine was the opposite, what is your explanation?

·         All tables and figures should be explained with throughout details.

-          Discussion:

·         This section should be rewritten once again by comparing with relevant literatures putting your explanation and possible causes, not only by mentioning the number of the references.

·         Remove the reference no. [39] as it is your result, and resort the references again within text and in references section.

·         The second part of the discussion made correlation between the AFM1 and the type of food consumption, as it was not the aim of the work, so you should collect samples from those food and test for AFM1 to be fair enough study.

·         All mentioned points above in result section by default should be in consequence appear and discussed here.

-          References:

·         Should be updated till 2023.

·         Add DOI to ref. whenever found.

·         Add all authors’ names instead of Et al.

Comments on the Quality of English Language

Extensive editing of English language required

Reviewer 2 Report

Comments and Suggestions for Authors

This paper addresses the measurement of aflatoxin metabolites in the urine of workers in Bangladesh. It holds significance due to the scarcity of data regarding aflatoxin contamination and exposure levels in Bangladesh. The study aimed to identify sources of exposure, particularly focusing on workers. However, a notable limitation is observed in the study design, as it does not account for crucial occupational exposure variables, including working hours, ventilation rates, and environmental characteristics of the workspaces. Therefore, I propose a major revision along with the following comments.

1. The primary objective of this study is to present the exposure levels of aflatoxin B1 in mill workers, comparing them with the general population(control group) to identify exposure levels and sources. However, the study predominantly presents correlations between food habits and exposure levels, without addressing the specificity of the working environment. Therefore, it is suggested to incorporate data on various occupational exposure variables for each worker, such as workspace volume, ventilation rate, working environment conditions (humidity, temperature, etc.), and working hours. Providing this data would enable the presentation of correlation between these occupational exposure factors and the levels of aflatoxin exposure.

2. The authors claim the frequent presence of AFM1 in the urine of mill workers(96.1%) and those of concurrent controls(92%) with dietary AFB1 exposure only. Is this statistically significant? If not, this statement should be removed.

3. Mill workers and the control group exhibit a significant difference in the only variable, BMI, where mill workers have BMI within the normal weight range (18.5-24.9), while the control group falls within the overweight range (25-29.9). It is recommended to provide an explanation for this discrepancy in the manuscript.

4. A formula for adjusting urine levels for creatinine needs to be provided.

(minor revision)

5. In Figure 1 caption, please change AM1 to AFM1

6. Change ml to mL throughout the manuscript.

Round 2

Reviewer 1 Report

Comments and Suggestions for Authors

Thanks for your editing and the manuscript now is more advanced than version no. 1, but there are still references in materials and methods section should be added in 5.1. Sampling so according to this reference you collected your samples, as well as in 5.3. Sample preparation and AFM1 analysis you should add reference according to which you perform your analysis.

Author Response

Response: We sincerely appreciate the time and effort you have put into further reviewing our manuscript. We have added in Section 5.3. a reference to one of our previous studies where we used ELISA for AFM1 analysis in the urines of an adult cohort in Rajshahi district of Bangladesh (Ali et al 2016, Arch Toxicol 2016, 90, 1749–1755, doi:10.1007/s00204-015-1601-y). In Methods section 5.1, we added also a reference [49] on creatinine adjustment of biomarker values (UBA 2005; doi:10.1007/s00103-005-1029-2).

Reviewer 2 Report

Comments and Suggestions for Authors

All comments are sincerely reviewed and authors made appropriate revisions to the manuscript. Therefore, I recommend accepting this manuscript in its present form.

Author Response

Response: Thank you indeed for the careful review, the very helpful comments and the positive feedback on the revisions made earlier in our manuscript as appropriate.